# Volatile Fingerprint Mediates Yeast-to-Mycelial Conversion in Two Strains of *Beauveria bassiana* Exhibiting Varied Virulence

**DOI:** 10.3390/jof9121135

**Published:** 2023-11-24

**Authors:** Arturo Ramírez-Ordorica, José Alberto Patiño-Medina, Víctor Meza-Carmen, Lourdes Macías-Rodríguez

**Affiliations:** Instituto de Investigaciones Químico Biológicas, Universidad Michoacana de San Nicolás de Hidalgo, Ciudad Universitaria, Morelia C.P. 58030, Michoacán, Mexico; arturo.ramirez.ordorica@umich.mx (A.R.-O.); jpatino@umich.mx (J.A.P.-M.); victor.meza@umich.mx (V.M.-C.)

**Keywords:** entomopathogen, dimorphism, environmental stimuli, volatiles, 3-methylbutanol, quorum sensing

## Abstract

*Beauveria bassiana* is a dimorphic and entomopathogenic fungus with different ecological roles in nature. In pathogenic fungi, yeast-to-mycelial conversion, which is controlled by environmental factors, is required for virulence. Here, we studied the effects of different stimuli on the morphology of two *B. bassiana* strains and compared the toxicities of culture filtrates. In addition, we explored the role of volatiles as quorum sensing-like signals during dimorphic transition. The killing assays in *Caenorhabditis elegans* (Nematoda: Rhabditidae) showed that strain AI2 isolated from a mycosed insect cadaver had higher toxicity than strain AS5 isolated from soil. Furthermore, AI2 showed earlier yeast-to-mycelial switching than AS5. However, an increase in inoculum size induced faster yeast-to-mycelium conversion in AS5 cells, suggesting a cell-density-dependent phenomenon. Gas chromatography-mass spectrometry (GC-MS) analyses showed that the fingerprint of the volatiles was strain-specific; however, during the morphological switching, an inverse relationship between the abundance of total terpenes and 3-methylbutanol was observed in both strains. Fungal exposure to 3-methylbutanol retarded the yeast-to-mycelium transition. Hence, this study provides evidence that volatile compounds are associated with critical events in the life cycle of *B. bassiana*.

## 1. Introduction

Dimorphic fungi have a unique attribute of transitioning between yeast and mycelial phases. Environmental stimuli such as temperature, pH, and nutrition influence this morphological change [1,2,3,4]. Several fungal pathogens of animals and plants are dimorphic, and the transition between the two phases favors fungal in vivo survival and pathogenesis [5]. Breakthroughs at the molecular level reveal the mechanisms underlaying the infection process in which virulence factors aid in pathogen colonization, either during the transition of mycelia to the parasitic phase or during host immunosupression [6,7,8]. For example, molecules such as branched-chain amino acids synchronize biological events related to the dimorphic transition in *Ceratocystis* (*Ophiostoma*) *ulmi*, the causal agent of Dutch Elm disease [1,9]. Likewise, hydroxy fatty acids stimulate filament formation in *Ustilago maydis* [10]. Moreover, certain volatile organic compounds (VOCs) such as alcohols have been reported to regulate fungal cell elongation, nuclear division, budding pattern, and conidiation [11,12,13]. For example, 3-methylbutanol promotes the continuous budding phenotype in *Saccharomyces cerevisiae*, and the aromatic alcohol tyrosol and sesquiterpene farnesol control physiological activities in the polymorphic opportunistic fungus *Candida albicans* [5,11,13,14,15,16].

*Beauveria bassiana* is a dimorphic fungus that infects arthropods of all major insect orders [17,18]. Furthermore, *B. bassiana* survives in the soil and can colonize different parts of the plant as endophytes [19,20]. Virulence among *B. bassiana* strains varies, however, and the physiological state of the insect host is an important factor for successful infection [21,22]. Usually, the fungal spores adhere to and germinate on the insect surface and penetrate the insect cuticle until they reach the hemocoel; here, the mycelium transforms to yeast-like cells called blastospores or hyphal bodies. During the last stage of the infection, blastospores transit to mycelia and emerge from the insect cadaver, producing newly infective conidia that are disseminated towards a new insect host, thus starting a new cycle of infection [18]. Recently, crucial activator genes such as *BbbrlA* and *Bbaba* of the central developmental pathway have been described as regulators of the fungal transition from filamentous form to blastospore and aerial conidiation [23]. However, the involvement of fungal quorum sensing-like molecules modulating dimorphic switching in *B. bassiana* remains unknown. Nevertheless, the mycotoxins produced, such as beauvericin, bassianolide, and oosporein, act simultaneously as diffusible chemical signals for fungal development, aggressiveness and/or virulence [8,18,22].

Indirect interactions between *B. bassiana* and its insect host can also occur via VOC emissions. Thus, prior to colonization, fungal VOCs modulate different types of behaviors, such as oviposition [8,24,25,26,27,28].

This study was based on the fact that the yeast-to-mycelial conversion in dimorphic fungi is affected by external factors and is important for pathogenesis and virulence. Here, we studied two strains of *B. bassiana* exhibiting varied virulence. One of the strains showed an early dimorphism, which was associated with an increased toxicity of culture filtrate. Additionally, we analyzed volatile molecules that regulate fungal morphogenesis via quorum sensing (QS). Our findings provide new information on the emission of volatiles during the fungal life cycle and add to our knowledge on the in vitro growth of *B. bassiana*.

## 2. Materials and Methods

### 2.1. Fungal Growth Conditions

The strains AI2 and AS5 of *B. bassiana* used in this study were previously isolated and identified by Ramírez-Ordorica et al. [28]. The fungal strains were grown and maintained on potato dextrose agar medium (PDA; BD Bioxon^®^, Mexico City, Mexico) in Petri dishes.

Yeast-like growth was observed on MacConkey agar medium (BD Bioxon^®^) [29]. To determine the effect of inoculum size on the yeast-to-mycelium conversion, the culture medium was inoculated with 100, 500, 10^3^, 10^6^, or 10^7^ spores using the spatulate method. Thereafter, the fungus was grown at room temperature (24 ± 2 °C) in the dark. Fungal development was observed under an optical microscope (Nikon^®^ Eclipse E200; Tokyo, Japan).

Liquid culture media were used to determine the effect of nutrition and pH on the spores’ germination at 28 °C. They included (1) YPG-2%, consisting of 20 g glucose per liter (Sigma^®^, St. Louis, MO, USA), 10 g gelatin peptone (Sigma^®^), and 3 g yeast extract (Sigma^®^); (2) YPG-6%, with a higher content of glucose (60 g per liter); (3) YNB minimal medium containing 6.7 g of YNB (Difco^®^, Mexico City, Mexico) and 20 g of glucose, to which, after sterilization, 10 µg/mL niacin (Sigma^®^) and thiamine (Sigma^®^) were added; and (4) PDB (potato dextrose broth) at pH 3 and 9. A conidial suspension (5 × 10^5^ conidia/mL) was used to produce a series of 125 mL Erlenmeyer flasks, each containing 10 mL of culture medium. The PDB culture medium at pH 6 was used as the standard. Aerobiosis was achieved with constant shaking at 150 rpm (14.4 ± 0.64% oxygen), and for anaerobiosis, we used a previously described system [30]. Fungal development was observed under an optical microscope (Olympus^®^ CKX41, Tokyo, Japan) equipped with a 40× objective lens and a DMC-T25 camera (Panasonic^®^, Kadoma, Japan).

### 2.2. Caenorhabditis Elegans Killing Assays

The nematode *C. elegans* Bristol N2 was kindly donated by Dr. Víctor Meza-Carmen. The nematode was grown to obtain worms in the young adult phase, as previously described [31]. The *B. bassiana* strains (5 × 10^5^ conidia/mL) were grown on PDB under aerobic conditions. Thereafter, the cell-free culture media were obtained after 48 h of growth, recovered via filtration (Millipore 0.2 µM filters; Billerica, MA, USA), and co-incubated with the worms for 72 h at 18 °C. For each experiment, 15 worms were dispensed into each well of a 24-well Costar plate (Corning, Inc., Armonk, NY, USA) with 1 mL of cell-free media. Sterile PDB culture medium was used as the negative control. Survival was monitored every 12 h, and the worms were considered dead when they no longer responded to the moving touch stimulus.

### 2.3. Fungal Volatile Analyses

The solid-phase microextraction technique (SPME, Supelco^®^, Bellefonte, PA, USA) was used for the extraction of volatiles from yeast-like growth of *B. bassiana*. Each fungal strain was inoculated into 4 mL SPME vials containing MacConkey slant medium (1.5 mL). The samples were kept in the dark at room temperature, and analyses were performed after 48, 72, 96, 120, and 144 h of fungal growth. The methodology used was the same as that used by Ramírez-Ordorica et al. [28]. A blue fiber (65 μm PDMS/DVB) was inserted into the vial for 30 min, and the compounds were desorbed in the injection port of a gas chromatograph (GC, Agilent 6850 Series II, Agilent, Foster City, CA, USA) coupled with a mass spectrometer (MS, Agilent 5973, Agilent, Foster City, CA, USA) at 180 °C for 30 s. A free fatty acid-phase capillary column (HP-FFAP) was used as the stationary phase. Analyses (n = 6) were programmed in the scan mode with an acquisition range of 45–250 *m*/*z*. All compounds were tentatively identified based on their best match to the NIST/EPA/NIH mass spectra database 11 and the NIST Mass Spectral Search Program 2.0 (ChemStation Agilent Technologies Rev. D.04.00.2002, Santa Clara, CA, USA). Compounds emitted only by the MacConkey agar medium were not considered.

For reference, the Kovats retention index was calculated using an alkane pool of C_5_-C_25_ and compared with that in the standard literature [32]. Finally, 3-methylbutanol was identified using a pure standard (>98.5% purity, Sigma-Aldrich^®^ I9392, St. Louis, MO, USA).

### 2.4. Effect of 3-Methylbutanol on Yeast-to-Mycelial Transition in B. bassiana

The strains of *B. bassiana* were grown on MacConkey agar medium on one-half of the I-plates, as described previously. In the other half of the Petri dish, a paper disk (3.5 cm) was placed with 9.5 μL of 3-methylbutanol (~7.8 μg). For the control conditions, only a paper disk was used. The fungal growth was monitored at different time points, i.e., 48, 72, 96, 120, and 144 h.

### 2.5. Statistical Analysis

Data obtained from fungal spore germination and development on different culture media were subjected to one-way ANOVA, followed by Fisher’s LSD post hoc test. To compare the survival of *C. elegans* populations exposed to the fungal secretome, we analyzed the survival curves by using the log-rank Mantel–Cox test.

The relative abundances of volatiles were analyzed via one-way ANOVA followed by Fisher’s LSD post hoc test using the R language version 4.1.2 (agricolae package) and STATISTICA 7.0.6. In addition, to compare the metabolic profiles of the volatiles, a principal component analysis (PCA) was performed using a normalized and centered abundance matrix of the compounds. Differences between treatments were assessed using PERMANOVA. Potential biomarkers were identified by taking the absolute values of the loading matrix from the first principal component and sorting them in descending order for both isolates (AI2 and AS5). Multivariate analysis was performed in R (version 4.1.2), using the vegan package.

## 3. Results

### 3.1. Nematicidal Activity of B. bassiana against C. elegans

A *C. elegans* model was used to compare the in vivo toxicities of culture filtrates from AI2 and AS5 strains (Figure 1). The nematodes were exposed to the cell-free culture media, and the survival rate was recorded over 72 h. Although the assay revealed that both strains reduced the survival rate of *C. elegans*, there were significant variations (*p* < 0.01) in toxicity between the two strains. The filtrate obtained from AI2 showed a higher toxicity, killing 60.0 ± 9.0% of the nematode population at the end of the assay, more than that (16.7 ± 4.6%) killed using the filtrate from AS5.

### 3.2. Effect of Growth Conditions on the Culture Morphology of B. bassiana Strains

The dimorphic fungus *B. bassiana* can develop yeast-like cells on MacConkey agar medium [29]. Fungal conidia (10^6^) of the AS5 and AI2 strains were inoculated into the culture medium, and the yeast forms were observed after 48 h of growth. Otherwise, the abundance of conidia differed between the strains, with the numbers of conidia reaching a maximum at 120 h in AI2. In contrast, there was no significant increase in counted conidia or the number of yeast-like cells in AS5 under similar assay conditions (Figure 2A,B).

The AS5 strain showed a slow development on the MacConkey agar medium, and the yeast phase prevailed throughout 144 h of growth (Figure 2B). As expected, other growing conditions modified the fungal morphology. In the case of the PDB culture medium, all conidia germinated as filamentous growths (Figure 3A,B). Meanwhile, on YPG-2%, conidial germination decreased by 20%, but the hyphae were longer than those observed on PDB. An increase in the carbon source concentration in YPG-6% and YNB culture media negatively affected conidial germination and hyphal growth. Furthermore, the anaerobic conditions negatively affected conidial germination and conidial germ tube length (Figure 3C).

The effect of pH on colonial morphology was also determined. The results showed that pH 6 was optimal for conidial germination and mycelial growth (Figure 4A,B). A pH of 3 significantly decreases conidial germination and limits hyphal growth, whereas a pH of 9 stimulates hyphal branching under aerobic conditions. Under anaerobic conditions, conidial germination and the morphology of mycelium were affected at pH 6 because the hyphae were more branched than those under aerobic conditions (Figure 4C). In addition, the fungal growth in a more acidic culture medium allowed for hyphal elongation, whereas under alkaline conditions, the hyphae were scarce and poorly branched.

### 3.3. Identification of Volatile Compounds during Yeast-to-Mycelial Phases in B. bassiana

Interestingly, the AS5 strain showed a slower development on the MacConkey agar medium than AI2 (Figure 5), and the yeast phase prevailed till 144 h, which was the maximum duration of the experiment (Figure 2). This observation suggests that the morphological change in *B. bassiana* may be associated with a cell-concentration-dependent phenomenon and that signaling molecules are required to induce the morphogenetic switch.

Inoculum sizes of 100, 500, and 1000 spores allowed for yeast-like development in AS5 at 6 d post-inoculation, whereas mycelial development was observed in AI2 (Figure 5A). The highest inoculum size of 10^7^ spores promoted dimorphic transition in both strains, and mycelial development was evident after 2 d of fungal growth (Figure 5B).

Volatile emission from both the strains was subjected to kinetic analyses using an inoculum size of 10^6^ spores. The results showed significant differences between strains and growth time (significant results, *p* ˂ 0.001, via permutational multivariate analysis of variance, PERMANOVA) (Figure 6A,B). The compounds that contributed the most to the projection of the data in the PCA loading plot are shown in Figure 6C,D. These compounds were primarily terpenes and the alcohol 3-methylbutanol, which showed the greatest variations throughout the 144 h of the experiment (Table 1 and Table 2).

The compounds were identified as alcohols, ketones, terpenes, or acids, the contents of which varied throughout the dimorphic transition of a fungus (Table 1 and Table 2). Compounds such as 1-octen-3-ol, 2-undecanone, α-selinene, guaia-9,11-diene, 10s,11s-himachala-3(12),4-diene, and α-gurjunene were specifically identified within the chromatographic profile of the volatiles produced by the AI2 strain, whereas (+)-2-bornanone, 2-methylisoborneol, 2-isopropenyl-4a,8-dimethyl-1,2,3,4,4a,5,6,7-octahydronaphthalene, 3,7(11)-selinadiene, and 2,4-diisopropenyl-1-methyl-1-vinylcyclohexane were identified within the profiles that corresponded to the AS5 strain. Only 3-methylbutanol and α-terpineol were identified in both strains and at all sampling times.

Interestingly, at 72 h of fungal growth, where the yeast phase was observed in both strains, the content of alcohols, ketones, and acids decreased, whereas a transient rise in terpenes was detected (Table 1 and Table 2). Nevertheless, 3-methylbutanol decreased at 72 h in AI2 (62.53%) and AS5 (16.63%) but increased consistently thereafter until reaching a concentration similar to that observed at 24 h (Table 1). Thus, the abundance of terpenes and 3-methylbutanol is a dynamic process in synchrony with the dimorphic transition and may act as a chemical biomarker in the interconversion of morphologies.

### 3.4. The Role of 3-Methylbutanol in the Dimorphism of B. bassiana

The exogenous addition of 3-methylbutanol to one-half of the I-plates affected the yeast-to-mycelium transition in the strains of *B. bassiana* (Figure 2A,B). In the case of AI2, which showed a precocious dimorphism in MacConkey agar medium, alcohol affected the spore quantity trade-off at 120 h of fungal growth. Furthermore, in AS5, where yeast growth prevailed in the MacConkey agar medium, alcohol promoted the yeast form and simultaneously increased the number of spores (Figure 7). Nevertheless, the spore count did not exceed that of the yeast cells. These results suggest that 3-methylbutanol exhibits activity as a single compound, perhaps by retarding the dimorphic transition of the fungus.

## 4. Discussion

*Beauveria bassiana* is a dimorphic and entomopathogenic fungus widely used for pest biocontrol [18]. Morphological transition in dimorphic fungi is crucial for pathogenicity but simultaneously defines its biology and prevalence in different environments [5]. In the present study, we examined the dimorphism of two *B. bassiana* strains previously isolated from different environmental niches. Strain AS5 was isolated from soil and AI2 from a mycosed insect cadaver. According to Ramirez-Ordorica et al. [28], AI2 was more lethal than AS5 when inoculated with conidia on larvae in stage L2 from the insect pest *Spodoptera frugiperda*. Notably, both strains killed the larvae, but the median lethal time was different. The results of the toxicity assays in *C. elegans* showed that the cell-free medium obtained from the aerobic growth of AI2 in PDB increased nematode killing compared with that observed with AS5, suggesting an important difference between the fungal secretomes that may induce mortality in *C. elegans* and *S. frugiperda*. The nematicidal effects of *B. bassiana* are widely known and different cyclic depsipeptides and proteins are directly related to this activity [33,34,35]. Hence, it would be interesting to perform a further analysis to reveal and compare the secreted metabolites of both strains to provide more information on the molecular players that participate in fungal colonization and virulence.

It is noted that any change in the insect host, carbon/nitrogen sources, inoculum size, or other types of stimuli could accelerate or slow down the life cycle of *B. bassiana* with direct repercussions on virulence [36,37]. In the case of AS5, both the culture media YPG and YNB, along with oxygen availability and pH, modified conidial germination and mycelium morphology, which are important morphological attributes for viability, colonization, and persistence in different environments [4,36,38,39,40,41,42]. *B. bassiana* thrives on a wide variety of substrates and pH ranges. AS5 showed low conidial germination at pH 3 and optimum mycelial growth at pH 6 on PDB medium under aerobic conditions. This is consistent with the results of previous studies [36,38,39]. Furthermore, pH 9 allowed for more branched mycelial growth. This modification in mycelial morphology may be the result of an adaptation to alkaline conditions [41]. This could represent an advantage for the saprophytic colonization of alkaline soils.

The relationship between morphological characteristics such as conidial germination speed, growth rate, and virulence capacity is controversial in *B. bassiana* [39,40,41,42,43,44]. Our observations using the MacConkey agar culture medium showed that the more virulent strain AI2 developed faster than AS5. Notably, AI2 switched faster between the two morphologic forms than AS5 did. Using an inoculum size of 10^6^ spores, a higher germination response and earlier dimorphism were observed in AI2 than in AS5, thus showing varying development with consequential differences in the timing of the dimorphic change. The speed of the dimorphic transition was dependent on the inoculum size. The highest inoculum size of 10^7^ enabled the transition of yeast cells to hyphal growth as early as 2 d post inoculation for both strains, whereas with a lower inoculum size of 1000 or less, only AI2 formed mycelia on day 6. These results suggest that the dimorphic transition speed is dependent on the threshold cell population density and the production of chemical signals that trigger yeast-to-mycelium conversion, as seen in other dimorphic fungi [2,3,5,45,46]. Furthermore, the speed in the transition of AI2 is a feature that could be associated with its increased virulence against *S. frugiperda*.

The QS system in entomopathogenic fungi has previously been reported in *Metarhizium rileyi* and *Ophiocordyceps sinensis* [46,47]. According to these studies, the production of QS signals occurs in the hemolymph in response to a quorum of hyphal bodies during the advanced stages of insect infection. In particular, the exogenous addition of N-acetyl glucosamine, a monomer of chitin found in the exoskeleton of insects and fungal cell walls, increased hyphal formation in *O. sinensis*, whereas other fungal QS signals previously described in *C. albicans*, such as farnesol and tyrosol, significantly affected the dimorphism of *O. sinensis* [47]. The promiscuity in the QS receptors towards multiple autoinducers can explain the microbial interspecies communication. However, the specific elicitors produced by entomopathogenic fungi remain unknown.

An important contribution of this study was the identification of volatile compounds that may act as chemical signals involved in the lifecycle of *B. bassiana*. For example, we identified 1-octen-3-ol as a part of the aromatic profile of AI2. This compound has been directly involved in conidiogenesis in filamentous fungi such as *Trichoderma atroviride* [48], exhibited toxicity against insects [49,50], and retarded seed germination and growth in *Arabidopsis* plants [51].

The variation in the contents of terpenes and 3-methylbutanol also caught our attention. Compound 3-methylbutanol was previously identified as the major compound produced by the mycelia of AI2 and AS5 (53.40 and 72.73% of all components, respectively) cultured in the PDA medium, followed by terpenes (11.06 and 2.89%, respectively) [28]. We observed that the terpene content decreased and alcohol content increased as the mycelial phase progressed. Interestingly, this ratio remained consistent regardless of virulence capacity, development, or source of isolation; nevertheless, more studies are needed to validate this observation.

Terpenes play various ecological roles [52]. Farnesol is a sesquiterpene alcohol that aids cell-to-cell communication, regulates important virulent traits such as morphogenic transition in *C. albicans* [53], and has shown similar activities in other dimorphic fungi as well [47]. *B. bassiana* emits different types of terpenes during the yeast phase. However, there are no reports on their participation in dimorphic transitions, despite the evidence of their biological activity in insects. For example, isomers of selinene, elemene, and guaiene have shown strong toxicity and antifeedant activity against the larvae of *Spodoptera* spp. [54,55,56,57].

The fusel alcohol 3-methylbutanol is a bioactive compound that acts as a messenger affecting the female egg-laying behavior in *S. frugiperda* [28], promoting the biomass accumulation in plants via auxin transport and signaling depending on the dose, application method, and plant age [58,59], and inducing filament formation in other fungi such as *Saccharomyces cerevisiae* and *C. albicans* [11,13,60]. The role of fusel alcohols as QS molecules has been previously reported for the dimorphic fungus *O. ulmi* and the polymorphic fungus *C. albicans* [9,15]. In *O. ulmi*, the authors identified 2-methylbutanol as the compound representing the major peak in the GC-MS profile, exhibiting activity as a QS molecule, thus influencing the fungal morphology. The compound 2-methylbutanol derives from isoleucine catabolism, which also affects the morphology. The dominant compound produced by *B. bassiana* is 3-methylbutanol, derived from leucine. We propose that 3-methylbutanol is part of a specific fingerprint of the compounds responsible for the dimorphic transition in *B. bassiana* and may be considered an autoregulatory QS signaling molecule in this fungus. The exogenous addition of the molecule to the I-plates showed activity as a QS molecule, retarding this phenomenon; the amino acid leucine in hemolymph probably acts as a trigger signal in *B. bassiana* because the effects of amino acids are associated with the synthesis of fusel alcohols [2]. The insect hemolymph is rich in free aminoacids [61,62]. Thus, the in vivo contribution of these endogenous components for fungal morphogenesis, propagation and disease establishment could be significant.

The effectiveness of *B. bassiana* in the development of infection and death in the host is the result of a combination of various factors [40,43,44]. To date, no correlation has been demonstrated between the source of isolation and virulence in *B. bassiana*. However, we observed that the isolate from the insect’s mycosed cadaver showed increased toxicity, sporulation speed, and earlier switching from the yeast to the mycelial form. This result is probably related to the life history traits of AI2. Thus, it is very important to conduct field experiments, as the observations obtained in an in vitro study may not always reflect the actual scenario in an agroecosystem. It also helps to define whether a fast-growing fungus with an early dimorphism is advantageous in biocontrol programs.

## 5. Conclusions

The life cycle of *B. bassiana* is marked by the dynamic emission of a pool of volatiles. Many of these have been reported to have biological activities in insects and plants, highlighting their multifaceted roles in nature. Importantly, volatiles affect morphogenesis in *B. bassiana*, and a temporal and specific code is required to regulate this phenomenon. The individual contribution of each identified compound to the dimorphic transition will need further attention; however, 3-methylbutanol acted as a QS signal. This finding represents a breakthrough in research on molecules that mediate QS activity in *B. bassiana*. The involvement of fusel alcohols as QS molecules during the establishment of infection should be explored in the future.

## Figures and Tables

**Figure 1 jof-09-01135-f001:**
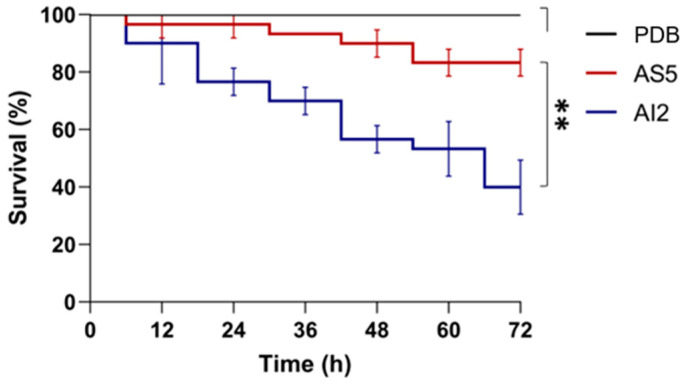
Survival rate of *C. elegans* reared in the presence of the secretome of *B. bassiana.* The fungal strains AI2 and AS5 were grown on PDB culture medium, and the cell-free media were used to perform the assays. A total of 15 nematodes were used per experiment and incubated at 18 °C for 72 h. n = 3, with three independent repetitions. Data were statistically analyzed using the Mantel–Cox test; **, *p* < 0.01.

**Figure 2 jof-09-01135-f002:**
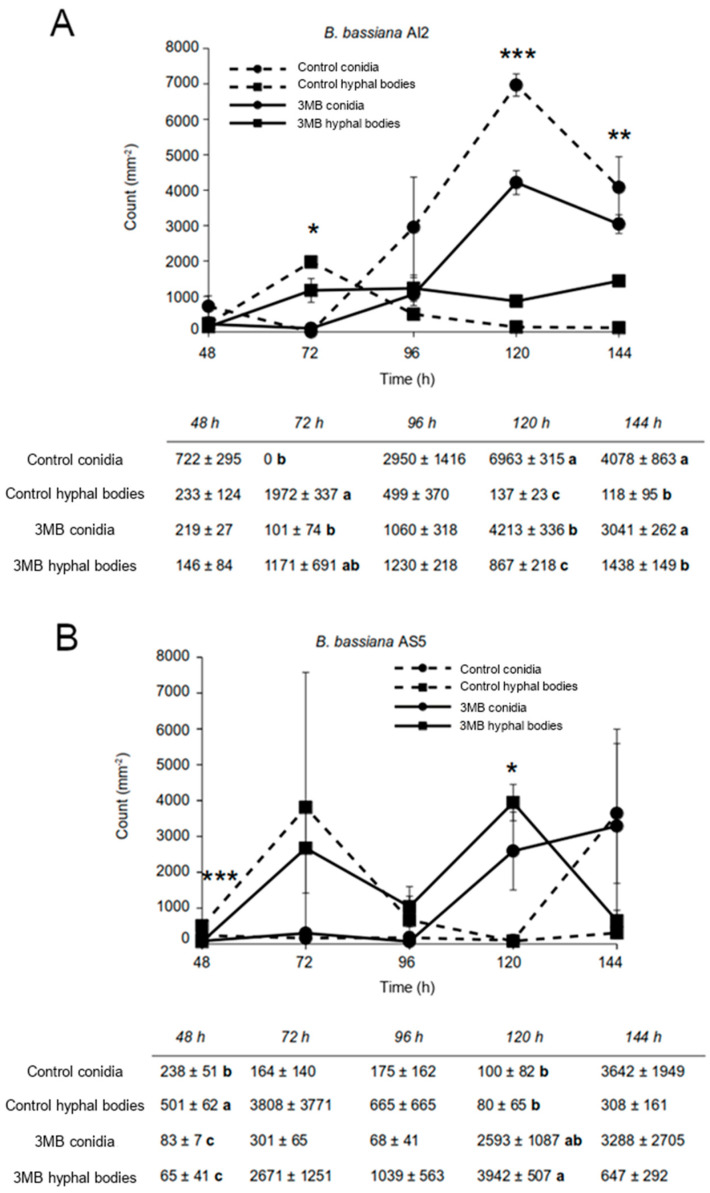
Development of *B. bassiana* AI2 (**A**) and AS5 (**B**) strains inoculated on MacConkey agar medium and exposed to 3-methylbutanol (3MB). The inoculum size was 10^6^ spores. The experiment was conducted in triplicate. Data are presented as mean ± standard error. Asterisks indicate means that differed significantly at * *p* ˂ 0.05, ** *p* ˂ 0.01, *** *p* ˂ 0.001 based on one-way ANOVA and Fisher’s LSD test. Means within columns with different letter indicate statistically significant differences using one-way ANOVA and Tukey’s test (*p* ≤ 0.05).

**Figure 3 jof-09-01135-f003:**
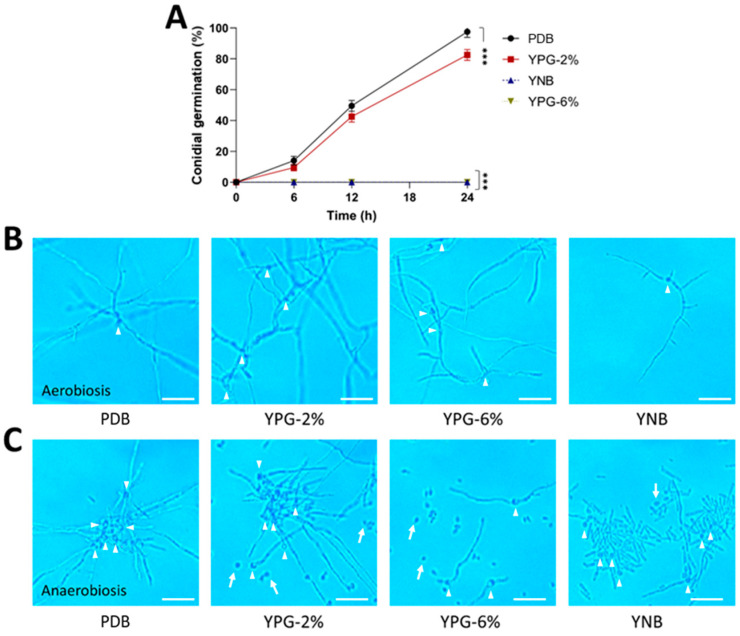
Effect of the carbon and nitrogen sources and oxygen on the morphology of the strain AS5 of *B. bassiana* at 24 h of fungal growth. Conidia were germinated in PDB, YPG-2%, YPG-6% and YNB for 24 h at 28 °C (anaerobiosis) and 150 rpm (aerobiosis). (**A**) Germination kinetics under aerobiosis conditions. Representative photographs at 100× of the morphology in (**B**) aerobiosis and (**C**) anaerobiosis at 24 h. White arrowheads indicate germinated conidia and white arrows indicate ungerminated conidia. Scale bar, 20 μm. n = 3, with three independent repetitions. Data were analyzed with one-way ANOVA and Fisher post hoc test (***, *p* < 0.001).

**Figure 4 jof-09-01135-f004:**
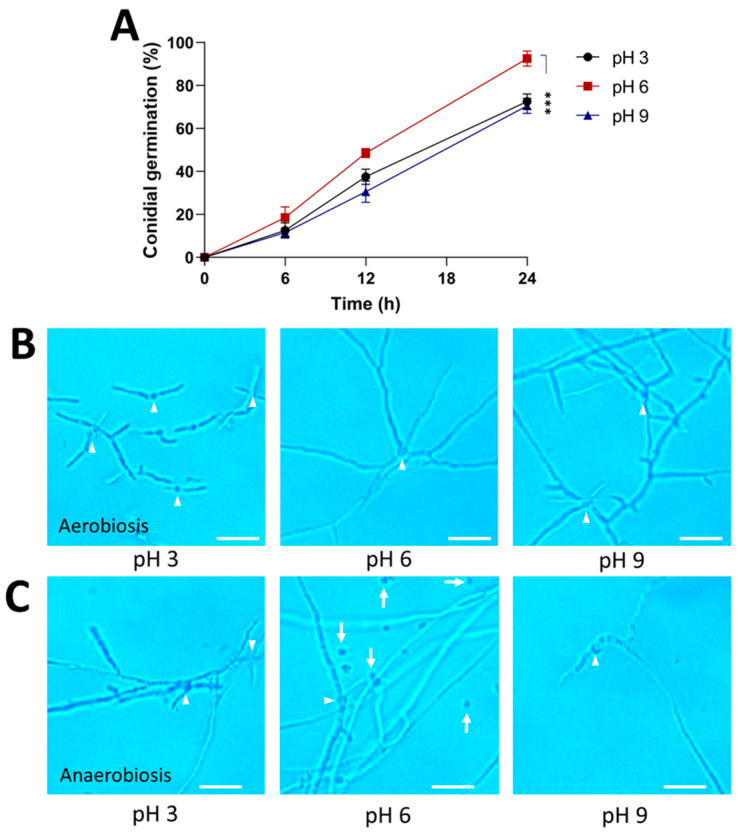
Effect of pH on the morphology of the strain AS5 from *B. bassiana*. Conidia were germinated in PDB at pH 3, 6, and 9 for 24 h at 28 °C (anaerobiosis) and 150 rpm (aerobiosis). (**A**) Germination kinetics under aerobiosis conditions. Representative photographs at 100× of the morphology during (**B**) aerobiosis and (**C**) anaerobiosis at 24 h. White arrowheads indicate germinated conidia and white arrows indicate ungerminated conidia. Scale bar, 20 μm. Data were analyzed with one-way ANOVA and Fisher post hoc test (***, *p* < 0.001).

**Figure 5 jof-09-01135-f005:**
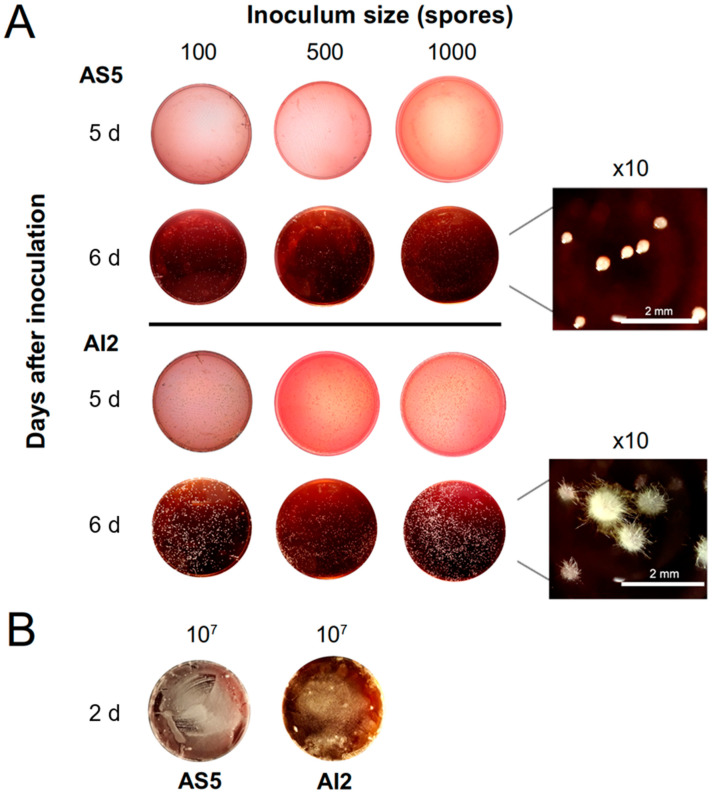
Effect of inoculum size on the morphology of *B. bassiana*. (**A**) Spores of the AS5 and AI2 strains (100, 500, and 1000) were inoculated in Petri dishes with 30 mL MacConkey agar medium. Representative photographs of the morphology for AS5 and AI2 at 5 and 6 d of the fungal growth. AS5 was developed as yeast cells in all inoculum sizes, whereas AI2 switched to mycelium phase on day 6. (**B**) Representative photographs of the mycelial growth for both strains on day 2 with an inoculum size of 10^7^ spores.

**Figure 6 jof-09-01135-f006:**
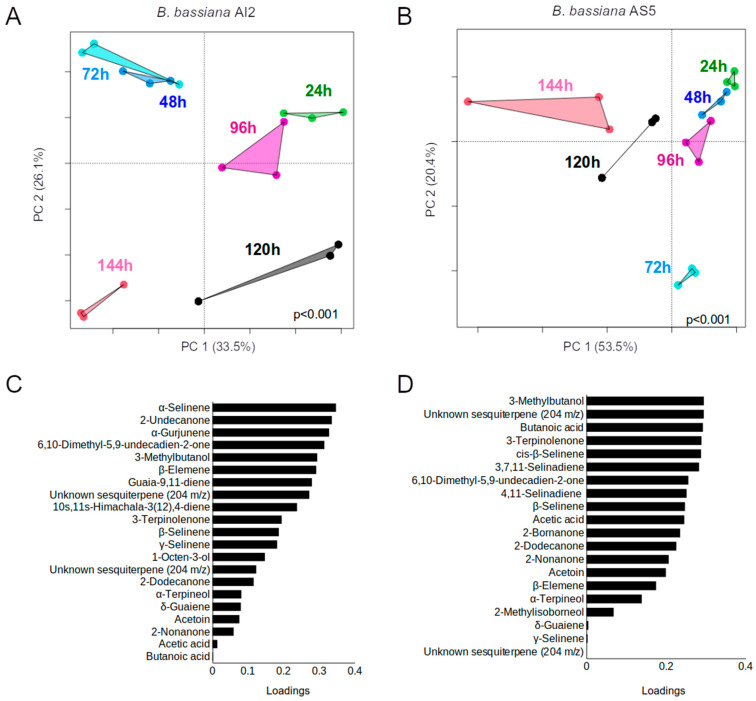
Principal component analysis (PCA) plots obtained based on the data of volatiles emitted by AI2 (**A**) and AS5 (**B**) strains on MacConkey agar medium and analyzed via GC-MS at different times of the fungal growth (n = 3). PERMANOVA (α = 0.05). (**C**,**D**) Plots depicting the contribution of the compounds to PCA using the relative peak areas (%) obtained in the chromatograms from AI2 and AS5, respectively.

**Figure 7 jof-09-01135-f007:**
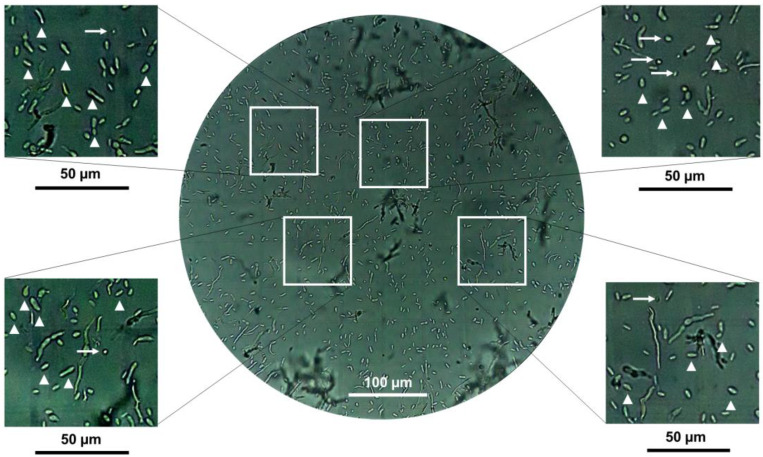
Representative microscopic image recorded at 40× for hyphal bodies, conidia, and mycelium of *B. bassiana* AS5 at 72 h of fungal growth and during exposure to 3-methylbutanol. An inoculum size of 10^6^ spores was inoculated on MacConkey agar culture medium. White arrowheads indicate yeast-like cells and white arrows indicate conidia.

**Table 1 jof-09-01135-t001:** Relative abundance (%) of volatiles identified in *B. bassiana* AI2 inoculated on MacConkey agar medium.

		Duration of Fungal Growth (h)					
Compounds	Ix	24	48	72	96	120	144
3-Methylbutanol	1186	70.57 ± 9.89 ab	91.56 ± 0.91 a	62.52 ± 4.01 b	61.03 ± 6.29 b	59.42 ± 5.66 b	78.99 ± 0.21 ab
1-Octen-3-ol	1451	2.97 ± 1.03 ns	0.91 ± 0.18 ns	nd	nd	nd	nd
Total alcohols		73.55 (8.86) ab	92.47 (0.72) a	62.52 (4.01) b	61.03 (6.29) b	59.42 (5.66) b	78.99 (0.21) ab
Acetoin	1285	nd	0.65 ± 0.04 b	0.32 ± 0.02 b	nd	0.35 ± 0.01 b	0.86 ± 0.03 ab
2-Nonanone	1389	9.31 ± 5.38 bc	0.90 ± 0.11 c	0.61 ± 0.06 c	4.59 ± 0.83 bc	6.01 ± 0.27 bc	4.99 ± 0.29 bc
2-Undecanone	1557	nd	nd	nd	nd	0.53 ± 0.26	nd
2-Dodecanone	1599	nd	nd	nd	1.60 ± 1.05 c	14.04 ± 1.23 ab	6.99 ± 0.95 bc
6,10-Dimethyl-5,9-undecadien-2-one	1850	nd	nd	nd	nd	0.18 ± 0.03 ns	nd
Total ketones		9.30 (5.38) de	1.55 (0.07) de	0.94 (0.091) e	6.19 (0.22) de	21.13 (1.20) bc	12.85 (0.71) cd
Acetic acid	1456	8.68 ± 1.55 a	1.50 ± 0.34 b	0.88 ± 0.19 b	nd	nd	nd
Butanoic acid	1633	3.79 ± 0.62 ns	1.01 ± 0.20 ns	0.43 ± 0.07 ns	nd	nd	nd
Total acids		12.47 (1.89) a	2.52 (0.36) b	1.31 (0.25) b	nd	nd	nd
β-Elemene	1508	nd	nd	nd	0.92 ± 0.08 ns	4.04 ± 3.14 ns	nd
Unknown sesquiterpene (m.w. 204)	1575	nd	nd	0.88 ± 0.10 b	0.71 ± 0.12 b	0.45 ± 0.08 b	nd
Unknown sesquiterpene (m.w. 204)	1586	nd	1.63 ± 0.29 e	13.88 ± 2.12 bc	13.03 ± 2.42 bc	5.00 ± 0.39 cde	3.54 ± 0.24 de
α-Selinene	1673	nd	nd	nd	0.78 ± 0.27 ns	1.45 ± 1.17 ns	nd
Guaia-9,11-diene	1678	nd	nd	1.24 ± 0.20 ns	0.39 ± 0.01 ns	1.68 ± 1.46 ns	nd
α-Terpineol	1697	4.67 ± 2.26 ab	1.17 ± 0.18 b	0.72 ± 0.15 b	0.52 ± 0.14 b	0.31 ± 0.09 b	0.28 ± 0.01 b
10s,11s-Himachala-3(12),4-diene	1704	nd	nd	0.39 ± 0.04 ns	0.32 ± 0.06 ns	0.24 ± 0.06 ns	nd
β-Selinene	1713	nd	nd	3.97 ± 0.52 ns	3.33 ± 0.71 ns	1.15 ± 0.12 ns	0.82 ± 0.02 ns
γ-Selinene	1718	nd	nd	5.46 ± 0.61 b	4.46 ± 0.93 b	1.62 ± 0.29 cd	1.09 ± 0.07 cd
α-Gurjunene	1733	nd	nd	nd	nd	0.46 ± 0.08	nd
δ-Guaiene	1757	nd	0.62 ± 0.02 d	7.33 ± 1.00 b	7.42 ± 1.51 b	2.20 ± 0.19 cd	1.48 ± 0.14 d
3-Terpinolenone	1923	nd	nd	1.29 ± 0.36 ns	0.82 ± 0.05 ns	0.78 ± 0.12 ns	0.90 ± 0.08 ns
Total terpenes		4.67 (2.26) d	3.43 (0.45) d	35.21 (4.20) b	32.76 (6.30) b	19.44 (5.66) bcd	8.14 (0.51) d

Compounds were analyzed using SPME-GC-MS and tentatively identified based on NIST library searches and Kovats retention indices (Ix). 3-Methylbutanol was confirmed by comparison with the standard. The mean values correspond to the sum of three independent determinations. Statistical analyses were performed for the individual compounds. Different letters indicate statistically significant differences according to a one-way ANOVA and Tukey’s test (*p* ≤ 0.05). Ix was calculated using a mixture of normal paraffin C_6_–C_20_ on an HP-FFAP capillary column and was compared with that available in the Pherobase database [30]. Rt: Retention time (min). nd: not detected. ns: not significant.

**Table 2 jof-09-01135-t002:** Relative abundance (%) of volatiles identified in *B. bassiana* AS5 inoculated on MacConkey agar medium.

		Duration of Fungal Growth (h)					
Compounds	Ix	24	48	72	96	120	144
3-Methylbutanol	1186	70.41 ± 8.06 ab	59.86 ± 5.15 b	16.63 ± 0.22 c	28.86 ± 3.22 c	64.96 ± 5.26 b	79.07 ± 1.75 ab
Total alcohols		70.41 (8.06) ab	59.86 (5.15) b	16.63 (0.22) c	28.86 (3.22) c	64.96 (5.26) b	79.07 (1.75) ab
Acetoin	1285	4.26 ± 1.64 a	2.46 ± 0.88 ab	0.45 ± 0.18 b	0.61 ± 0.02 b	2.64 ± 0.04 ab	2.50 ± 1.15 ab
2-Nonanone	1389	0.47 ± 0.08 c	2.64 ± 0.51 bc	0.41 ± 0.05 c	20.81 ± 4.44 a	12.34 ± 0.66 ab	nd
2-Dodecanone	1599	nd	nd	nd	18.54 ± 2.75 a	9.93 ± 2.70 abc	nd
6,10-Dimethyl-5,9-undecadien-2-one	1850	nd	nd	nd	0.51 ± 0.05 ns	1.76 ± 1.40 ns	nd
Total ketones		4.74 (1.61) de	5.10 (1.39) de	0.86 (0.15) e	40.49 (3.30) a	26.68 (3.41) b	2.50 (1.15) de
Acetic acid	1456	8.04 ± 2.13 a	5.03 ± 0.71 ab	0.66 ± 0.23 b	0.67 ± 0.17 b	0.39 ± 0.04 b	nd
Butanoic acid	1633	5.96 ± 2.96 ns	nd	nd	nd	nd	nd
Total acids		14.00 (3.90) a	5.03 (0.71) b	0.66 (0.23) b	0.67 (0.17) b	0.39 (0.04) b	nd
(+)-2-Bornanone	1383	nd	nd	nd	nd	0.57 ± 0.26	nd
β-Elemene	1508	nd	nd	nd	1.38 ± 0.37 ns	nd	2.12 ± 0.83 ns
Unknown sesquiterpene (m.w. 204)	1575	nd	nd	2.00 ± 0.20 a	1.08 ± 0.16 b	nd	nd
Unknown sesquiterpene (m.w. 204)	1586	nd	14.17 ± 2.52 b	43.86 ± 0.51 a	12.46 ± 3.65 bcd	3.02 ± 0.31 e	3.49 ± 1.58 de
2-Methylisoborneol	1592	nd	nd	nd	nd	nd	0.77 ± 0.19
2-Isopropenyl-4a,8-dimethyl-1,2,3,4,4a,5,6,7-octahydronaphthalene	1625	nd	nd	0.89 ± 0.26 ns	nd	0.66 ± 0.27 ns	nd
α-Terpineol	1697	10.83 ± 5.25 a	4.24 ± 0.23 ab	1.24 ± 0.18 b	1.87 ± 0.72 b	0.82 ± 0.12 b	0.66 ± 0.10 b
β-Selinene	1713	nd	2.84 ± 0.03 ns	6.08 ± 0.11 ns	2.02 ± 0.48 ns	0.77 ± 0.25 ns	9.52 ± 5.62 ns
γ-Selinene	1718	nd	3.53 ± 0.39 bc	8.42 ± 0.25 a	3.33 ± 0.91 bc	0.88 ± 0.45 cd	0.74 ± 0.33 d
3,7(11)-Selinadiene	1726	nd	nd	0.47 ± 0.08 ns	1.14 ± 0.62 ns	nd	nd
2,4-Diisopropenyl-1-methyl-1-vinylcyclohexane	1752	nd	nd	0.69 ± 0.12 b	3.83 ± 0.47 a	nd	nd
δ-Guaiene	1757	nd	5.19 ± 0.86 bc	18.13 ± 0.26 a	nd	1.20 ± 0.30 d	1.09 ± 0.40 d
3-Terpinolenone	1923	nd	nd	nd	2.82 ± 1.72 ns	nd	nd
Total terpenes		10.83 (5.52) cd	29.99 (3.07) bc	81.83 (0.57) a	29.96 (6.17) bc	7.95 (1.92) d	18.42 (2.62) bcd

Compounds were analyzed using SPME-GC-MS and tentatively identified based on NIST library searches and Kovats retention indices (Ix). The presence of 3-methylbutanol was confirmed by comparison with the standard. The mean values correspond to the sum of three independent determinations. Statistical analyses were performed for the individual compounds. Different letters indicate statistically significant differences according to one-way ANOVA and Tukey’s test (*p* ≤ 0.05). Ix was calculated using a mixture of normal paraffin C_6_–C_20_ on an HP-FFAP capillary column and was compared with that available in the Pherobase database [30]. Rt: Retention time (min). nd: not detected. ns: not significant.

## Data Availability

Data are contained within the article.

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
