# Peer review of "Volatile Fingerprint Mediates Yeast-to-Mycelial Conversion in Two Strains of Beauveria bassiana Exhibiting Varied Virulence"

_jof, 2023, doi:10.3390/jof9121135_

Round 1
Reviewer 1 Report
Comments and Suggestions for Authors
This manuscript presents experimental data on how the volatile alcohol 3-methylbutanol generated by two B. bassiana strains mediates their dimorphic transition, as claimed by the authors. Although their data are interesting and potential for use in improving mass production technology of B. bassiana on solid substrata, their interpretation of those data are quite questionable. I speculate that they were studying microconidiation on agar plates supplemented with special chemical rather than dimorphic transition, which can be readily observed for fungal insect pathogens grown in submerged liquid cultures or in hemolymph samples taken from insects surviving fungal infection. The confused interpretation can be easily clarified by adding microscopic images of spores produced under different culture conditions. Therefore, I suggest the authors to reorganize their data and rewrite the manuscript for submission.
Suggestions for authors’ consideration:
1. L150-151, the difference in survival rate is ascribed to the supernatants containing substances secreted by the two B. bassiana strains rather than their difference in virulence. I cannot see any relationship of such data with the main topic of the manuscript.
2. In my opinion, Figure 2 shows the observations of microconidiation on the agar plates rather than the yeast-like growth, which can be best observed in a broth such as trehalose-peptone broth mimicking insect hemolymph or in hemolymph samples taken from surviving insects after infection. Microscopic images of those spores produced in different treatments would help to judge this speculation. Why don’t you use a log-scale to distinguish the spore yields of different treatments in the colorful symbol-line charts with no need to repeat the data in the tables? Slow or fast growth of either strain can be shown with biomass levels on cellophane-overlaid agar plates rather than with the counts of spores.
3. Figure 3 does show a status of yeast-like budding proliferation in different liquid media, which should be better defined, such as potato dextrose broth (PDB) for agar-free PDA. Did you prepare the cultures by shaking incubation? If so, don’t use the term anaerobiosis to describe the culture condition!
4. In Figures 3A and 4A, germination percentages are not an index of growth. A 24-h incubation of B. bassiana conidia at an optimal temperature often results in branched hyphal growth. How could you distinguish germinated and non-germinated conidia at the end of 24-h incubation?
5. Figure 6 presents no microscopic images for the status of ‘yeast-like growth’, reinforcing my speculation on whether you were studying dimorphic transition or microconidiation of the B. bassiana strains. The alcohol 3-methylbutanol revealed in the study could be an important factor to promote microconidiation. The yeast-like budding proliferation can be readily shown in submerged liquid cultures rather than on aerial agar plates.
6. By the way, dimorphic transition of either B. bassiana (https://doi.org/10.1128/mSystems.00140-19) or Metarhizium robertsii (https://doi.org/10.1128/spectrum.00070-23) is genetically controlled by the key asexual developmental activator BrlA or AbaA because either aerial conidiation or submerged blastospore production was abolished in the absence of brlA or abaA.
Comments on the Quality of English LanguageThe English writing is understandable and must be improved for clarity, conciseness and readability
Author Response
Reviewer 1:
This manuscript presents experimental data on how the volatile alcohol 3-methylbutanol generated by two B. bassiana strains mediates their dimorphic transition, as claimed by the authors. Although their data are interesting and potential for use in improving mass production technology of B. bassiana on solid substrata, their interpretation of those data are quite questionable. I speculate that they were studying microconidiation on agar plates supplemented with special chemical rather than dimorphic transition, which can be readily observed for fungal insect pathogens grown in submerged liquid cultures or in hemolymph samples taken from insects surviving fungal infection. The confused interpretation can be easily clarified by adding microscopic images of spores produced under different culture conditions. Therefore, I suggest the authors to reorganize their data and rewrite the manuscript for submission.
Dear Reviewer, We very much appreciate the constructive criticisms offered by you. The main goal of our study was to determine the role of fungal volatiles on the dimorphic transition of B. bassiana. Previously, Alves et al. (2002) (doi: 10.1016/s0022-2011(02)00147-7) showed that in this solid substrata, B. bassiana is able to develop yeast-like phase. We confirmed this finding and we used MacConkey culture medium to study the dimorphism in B. bassiana. Now, in the revised version of our manuscript, we include a new figure (Fig. 7) with a representative microscopic image of yeast cells and spores obtained on MacConkey culture medium.
It would be very interesting to study the role of 3-methylbutanol in microconidiation, but, it will require an appropriate experimental design. However, we believe that we show an important evidence of the role of volatiles and 3-methyl butanol as chemical signals modulating fungal development.
Now, I will proceed to discuss point by point your suggestions and the corresponding changes we made in the manuscript.
Suggestions for authors’ consideration:
- L150-151, the difference in survival rate is ascribed to the supernatants containing substances secreted by the two bassianastrains rather than their difference in virulence.
Agree. We apologize for this mistake. We modify the manuscript accordingly. Please, see L160-161. We changed the lines as follows:
“Although the assay revealed that both strains reduced the survival rate of C. elegans, there were significant variations (P < 0.01) in the toxicity between the two strains”.
I cannot see any relationship of such data with the main topic of the manuscript.
In the revised version of our manuscript we developed a better argument in support of these data with the main topic. Now, this part of the manuscript is more readability and highlights the novelty of the study.
We edited L69-72 as follows: “Here, we studied two strains of B. bassiana exhibiting varied virulence. One of the strains showed an early dimorphism which was associated to an increased toxicity of culture filtrate”.
- In my opinion, Figure 2 shows the observations of microconidiation on the agar plates rather than the yeast-like growth, which can be best observed in a broth such as trehalose-peptone broth mimicking insect hemolymph or in hemolymph samples taken from surviving insects after infection. Microscopic images of those spores produced in different treatments would help to judge this speculation.
Regarding to this comment, we add a new figure (Fig. 7) which contains a representative microscopic image of the yeast-like growth developed by B. bassiana on MacConkey agar medium. In this image, both, hyphal bodies and spores can be observed. Furthermore, budding-yeasts can be seen as evidence of yeast-like phase.
Why don’t you use a log-scale to distinguish the spore yields of different treatments in the colorful symbol-line charts with no need to repeat the data in the tables?
Thank you for the suggestion. However, we believe that in this case, the linear scale provides better data visualization.
Slow or fast growth of either strain can be shown with biomass levels on cellophane-overlaid agar plates rather than with the counts of spores.
Agree. We were more cautious in writing and we modified the manuscript accordingly.
- Figure 3 does show a status of yeast-like budding proliferation in different liquid media, which should be better defined, such as potato dextrose broth (PDB) for agar-free PDA.
Figure 3 corresponds to germinated and ungerminated conidia in different culture media. In the revised version we indicate with arrowheads and arrows these structures within the microscopic images. Furthermore, the figure caption was edited to avoid confusion. In addition we replace agar-free PDA with PDB.
Did you prepare the cultures by shaking incubation? If so, don’t use the term anaerobiosis to describe the culture condition!
Concerning to this comment, we edited the figure caption to avoid confusion.
- In Figures 3A and 4A, germination percentages are not an index of growth. A 24-h incubation of B. bassiana conidia at an optimal temperature often results in branched hyphal growth. How could you distinguish germinated and non-germinated conidia at the end of 24-h incubation?
In the revised version of the manuscript we indicated germinated and ungerminated conidia with white arrowheads and arrows, respectively. We hope that this action serves to clarify what we refer to as germinated conidia.
- Figure 6 presents no microscopic images for the status of ‘yeast-like growth’, reinforcing my speculation on whether you were studying dimorphic transition or microconidiation of the B. bassiana strains. The alcohol 3-methylbutanol revealed in the study could be an important factor to promote microconidiation. The yeast-like budding proliferation can be readily shown in submerged liquid cultures rather than on aerial agar plates.
As I mentioned above, in the revised manuscript, a representative microscopic image of the yeast-like growth of B. bassiana on MacConkey agar culture medium was included (Fig. 7). Therefore, we are really observing the fungal dimorphic transition. It would be interesting to study the role of 3-methylbutanol in microconidiation, however it will require a different experimental approach.
- By the way, dimorphic transition of either B. bassiana(https://doi.org/10.1128/mSystems.00140-19) or Metarhizium robertsii (https://doi.org/10.1128/spectrum.00070-23) is genetically controlled by the key asexual developmental activator BrlA or AbaA because either aerial conidiation or submerged blastospore production was abolished in the absence of brlA or abaA.
Thank you for the comment. We know these studies. However, we are interested in the chemical signals that modulate the fungal growth and development. Within the hemocoel, the current model speculates that blastospores-to-mycelium conversion is initiated by some unknown signal. Such a signal may be depletion of critical nutrients found in the hemolymph or even production of a quorum-like sensing molecule by the fungus, although to date no such molecule has been identified. Our results provide information related to the timing of the transition and the presence of 3-metilbutanol suggests that the alcohol may act as QS autoregulator signal. In the revised manuscript the reference of Zhang et al. 2019 (https://doi.org/10.1128/mSystems.00140-19) was included. In addition in the manuscript was highlighted our interest to perform this study. Please see L56-64.
Reviewer 2 Report
Comments and Suggestions for Authors
Beauveria bassiana is an important pathogen used to control insect pests. The authors studied the yeast-to-mycelial conversion in two strains of Beauveria bassiana and described the volatile compounds in the life cycle of B. bassiana. In general
The manuscript was well-written and the experimental design looks good. The results were well presented. I provided several comments on the manuscript.

Author Response
Thanks for your comments. All the comments were addressed. Please see L17, 48, 102, and 113.
Round 2
Reviewer 1 Report
Comments and Suggestions for Authors
This manuscript has been revised following my previous comments and suggestions. I am satisfied with authors' revision in response to my comments and have no more questions.
Author Response
Reviewer 1:
This manuscript has been revised following my previous comments and suggestions. I am satisfied with authors' revision in response to my comments and have no more questions.
Thank you for the careful review of our manuscript.